# Broadband multi-magnon relaxometry using a quantum spin sensor for high frequency ferromagnetic dynamics sensing

Brendan A. McCullian [1]✉, Ahmed M. Thabt[1], Benjamin A. Gray[2], Alex L. Melendez[1], Michael S. Wolf[2], Vladimir L. Safonov[2], Denis V. Pelekhov[1], Vidya P. Bhallamudi [3], Michael R. Page [2] & P. Chris Hammel [1]✉

Development of sensitive local probes of magnon dynamics is essential to further understand the physical processes that govern magnon generation, propagation, scattering, and relaxation. Quantum spin sensors like the NV center in diamond have long spin lifetimes and their relaxation can be used to sense magnetic field noise at gigahertz frequencies. Thus far, NV sensing of ferromagnetic dynamics has been constrained to the case where the NV spin is resonant with a magnon mode in the sample meaning that the NV frequency provides an upper bound to detection. In this work we demonstrate ensemble NV detection of spinwaves generated via a nonlinear instability process where spinwaves of nonzero wavevector are parametrically driven by a high amplitude microwave field. NV relaxation caused by these driven spinwaves can be divided into two regimes; one- and multi-magnon NV relaxometry. In the one-magnon NV relaxometry regime the driven spinwave frequency is below the NV frequencies. The driven spinwave undergoes four-magnon scattering resulting in an increase in the population of magnons which are frequency matched to the NVs. The dipole magnetic fields of the NV-resonant magnons couple to and relax nearby NV spins. The amplitude of the NV relaxation increases with the wavevector of the driven spinwave mode which we are able to vary up to $3 \times 10^6 \, m^{-1}$, well into the part of the spinwave spectrum dominated by the exchange interaction. Increasing the strength of the applied magnetic field brings all spinwave modes to higher frequencies than the NV frequencies. We find that the NVs are relaxed by the driven spinwave instability despite the absence of any individual NV-resonant magnons, suggesting that multiple magnons participate in creating magnetic field noise below the ferromagnetic gap frequency which causes NV spin relaxation.

---

[1] Department of Physics, The Ohio State University, Columbus, OH 43210, USA. [2] Materials and Manufacturing Directorate, Air Force Research Laboratory, Wright-Patterson AFB, OH 45433, USA. [3] Department of Physics, Indian Institute of Technology, Madras, Chennai 600 036, India. ✉email: mccullian.1@osu.edu; hammel.7@osu.edu

Magnons, the quanta of spinwaves, are central to the development of modern spintronics[1]. In applied settings magnons have been shown to transmit spin angular momentum over large distances[2], can be emitted locally by spin-torque oscillators[3], can be used to coherently manipulate defect-based spin qubits[4,5], and can form the elements of spin-based transistors[6]. Key to the development of magnon applications is the parallel development of magnon sensing techniques which can elucidate the physical phenomena that govern magnon generation, propagation, scattering, and relaxation.

Quantum spin sensors, in particular the nitrogen-vacancy (NV) center defect spin in diamond, are ideal platforms for high sensitivity nanoscale probing of magnons. NV centers have a spin-dependent fluorescence intensity and long spin lifetime[7–9], which can be used to sensitively measure magnetic field noise at the NV resonance frequency which causes NV spin relaxation. Noise sensing with NVs, also called relaxometry, is a widely popular technique, which has been used to detect electron-phonon instability in graphene[10], spin labels[11–16], driven electron paramagnetic resonance[17,18], and ferromagnetic dynamics[19–26], among others. Early NV relaxometry studies of ferromagnetic resonance (FMR) provide a clear picture of the sensing scheme:[23] microwave drive excites a mode in a ferromagnetic film, the excited mode undergoes incoherent four-magnon scattering processes leading to a redistribution of magnon population throughout the magnon spectrum. This results in an increase in the number of NV-resonant magnons, which produce dipole magnetic field noise at the NV frequency and increase the relaxation rate of nearby NV spins. Thus far, however, NV relaxometry has had an inherent frequency limitation, set by the few GHz resonance frequency of the NV spin. The measurement scheme relies on the presence of an NV-resonant magnon mode which couples to and relaxes the NV spin. For the magnonics community, the frequencies of interest are often higher than NV frequencies[2], limiting the applicability of NV sensing.

Here we use NV relaxometry to study magnetic noise from magnons in a thin film of the insulating magnet nickel zinc aluminum ferrite (NZAFO). NZAFO is a low-damping ferrimagnetic insulator with large magnetic anisotropy which allows us to tune the frequency of the uniform FMR mode relative to the NV resonant frequencies with modest applied magnetic fields. Using a nonlinear spinwave generation process we can tune the microwave-excited spinwave wavevector by controlling the magnetic field and microwave power applied to the NZAFO. Varying the driven wavevector up to $3 \times 10^6\,\mathrm{m}^{-1}$, well into the exchange magnon spectrum for NZAFO, we find that the magnetic field noise detected by the NV ensemble increases with increasing wavevector until we reach a sharp cutoff at the critical field for the nonlinear spinwave generation process. We then increase the magnetic field such that all spinwave modes are at higher frequency than the NV frequencies, finding that the nonlinearly driven NZAFO magnetization still results in NV relaxation. This first demonstration of broadband multi-magnon NV relaxometry is a promising result for high-frequency magnetic dynamics sensing with NVs.

For clarity, we explicitly define some terms used throughout. Single magnon NV relaxometry refers to the process of a single magnon at the NV frequency whose dipole magnetic field couples to and relaxes nearby NV spins. Multi-magnon NV relaxometry refers to the process where multiple magnons participate to create NV-resonant dipole magnetic field noise which relaxes NV spins. Second-order spinwave instability is a nonlinear spinwave generating process whereby a ferromagnetic film driven with a sufficiently large microwave field near to but not on the uniform mode resonance condition will be driven into resonance, resulting in pairs of counterpropagating nonzero-$\vec{k}$ spinwaves at the microwave drive frequency. Four-magnon scattering is a magnon-magnon scattering process within the ferromagnetic film which redistributes magnon population from the microwave-driven ferromagnetic mode throughout the magnon states.

## Results

Our device geometry is shown in Fig. 1a. The 23 nm thick NZAFO ($Ni_{0.65}Zn_{0.35}Al_{0.8}Fe_{1.2}O_4$) film was grown on a $MgAl_2O_4$ substrate using pulsed laser epitaxy[27]. A 15-μm wide tapered microstrip antenna composed of Ti(5 nm)/Ag(285 nm)/Au(10 nm) was fabricated on the NZAFO film along the NZAFO 100 crystalline axis and was used to excite ferromagnetic dynamics. An electromagnet creates an in-plane static magnetic field along the NZAFO 100 crystalline axis and is swept during measurement. A powder of roughly 100 nm diameter nanodiamonds with ~30 parts-per-million NV center concentration was drop-cast on both the stripline and NZAFO surfaces into a nanodiamond layer a few hundred nm thick. The nanodiamonds are randomly oriented with respect to the static field. A constant, 532 nm laser at 30 mW power was focused on a few μm wide spot through a 20× microscope objective to address the ensemble of NV spins. Red NV photoluminescence (PL) is collected through the same objective. All measurements presented were taken on the nanodiamond powder deposited on the stripline, giving an NV-NZAFO separation of roughly 300–600 nm. Additional

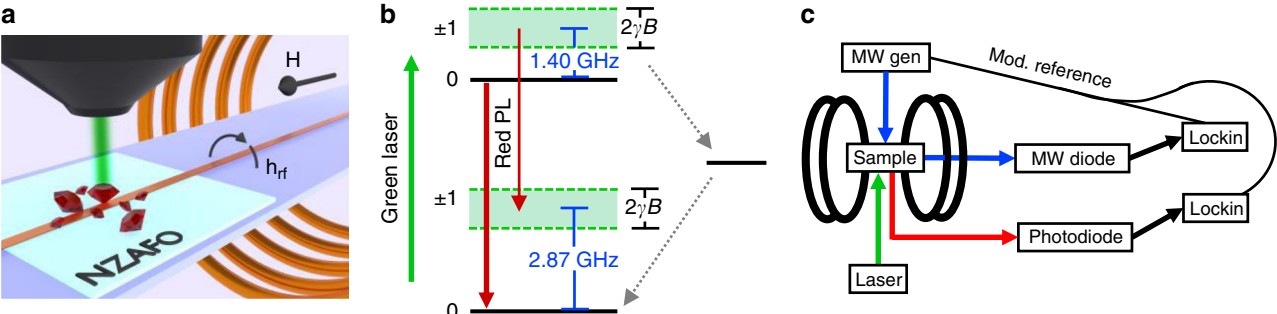

**Fig. 1 Experimental scheme. a** In-plane static magnetic field is applied and swept along the NZAFO thin film 100 crystalline axis. Microwaves from microstrip antenna drive NZAFO magnetization. Microscope objective focuses continuous green laser onto NV nanodiamond powder and collects red NV fluorescence. **b** NV energy diagram. The $|m_s = 0\rangle$ state has higher fluorescence intensity than the $|m_s = \pm 1\rangle$ states. Zeeman interaction with the static magnetic field splits the $|m_s = \pm 1\rangle$ states. For our experiment, the powder of randomly oriented nanodiamonds leads to a distribution of NV frequencies across the bands indicated in green. **c** Detection schematic. Amplitude modulated microwaves are applied to the sample, NV fluorescence and microwave transmission are recorded by lockin amplifiers referenced to this modulation during magnetic field sweeps.

measurements taken on nanodiamonds deposited directly on NZAFO were qualitatively similar to the results reported here.

We leverage the optical polarization and spin-dependent PL intensity of NVs to detect magnetic field noise at the NV spin transition frequencies. The NV center is a spin 1 defect and the ground state is a spin triplet as shown in Fig. 1b. Under laser excitation the $m_s = 0$ spin state has a higher PL intensity than the $m_s = \pm 1$ spin states due to the $m_s = \pm 1$ having higher probability of relaxing via a nonradiative pathway back to the ground $m_s = 0$ spin state. This nonradiative pathway is also responsible for optical polarization of the NV spins to a maximum PL $m_s = 0$ state under laser illumination. The PL intensity for a constant laser intensity is dependent on the NV transition rates, and thus if the relaxation rate of the $m_s = 0$ state increases the PL will decrease, which we can attribute to magnetic field noise at the NV $m_s = 0$ to $m_s = \pm 1$ transition frequencies. The ground state NV splitting is 2.87 GHz and the $m_s = \pm 1$ further split due to a Zeeman interaction with static magnetic field applied along the NV axis. Since we use a powder of randomly oriented nanodiamonds our NV spin frequencies will be determined by the strength of the static field and the distribution of angles between the NV axes and the static field. Our measurement technique is given in the Methods section.

At high microwave powers ferromagnetic samples undergo nonlinear response to microwave drive due to the parametric excitation of nonzero wavevector spinwaves[28–30]. For our transverse pumping geometry, we are able to drive the NZAFO film to a second-order spinwave instability process, called resonance saturation. At fields near the uniform mode ferromagnetic resonance (FMR) condition two microwave photons excite two uniform mode magnons at the microwave drive frequency $\omega_{MW}$, which then combine and produce two counter-propagating magnons of nonzero wavevector, also at $\omega_{MW}$[31]. Instability refers to the fact that beyond a certain microwave power threshold the rate of energy loss of these parametrically excited nonzero wavevector spinwaves can be compensated by the energy flow from the microwave drive and the spinwave mode will become excited above thermal levels[32].

A schematic of the microwave response of our ferrite sample for four different regimes of magnetic field is given in Fig. 2. For magnetic fields sufficiently below the uniform mode FMR condition, as in the case of Fig. 2a, microwaves do not efficiently couple to the uniform FMR mode and the microwave power needed to drive the instability will be prohibitively large. As the magnetic field is increased toward the uniform mode FMR condition, in Fig. 2b, the microwave power threshold to excite the spinwave instability is lowered and the resonance saturation process described above will populate nonzero wavevector spinwaves at $\omega_{MW}$. The spinwave branch most efficiently excited is $\vec{k} \parallel \vec{M}$[31]. As the magnetic field is further increased we eventually reach the conventional uniform mode FMR excitation of Fig. 2c. Finally, at higher magnetic fields there are no spinwave states at $\omega_{MW}$ and so no instability occurs. The process given in Fig. 2b is microwave power dependent, and is responsible for the evolution of the lineshape of the microwave absorption signal as a function of power.

Simultaneously acquired microwave absorption and NV response to the NZAFO at 2.2 GHz at +3 dBm microwave power are given in Fig. 3a, b, respectively. At this frequency and range of magnetic fields the microwave frequency is sufficiently far from all NV resonance frequencies that there is no direct microwave-NV coupling. The microwave absorption profile here shows the expected response during a field sweep at constant frequency as outlined in Fig. 2, with the uniform mode FMR signal at 94 Gauss and an absorption shoulder at a lower magnetic field. The cutoff of the absorption shoulder at 65 Gauss corresponds to the critical microwave field for the instability process, and will produce the largest wavevector spinwave at this microwave frequency and power. The NV PL signal shows that the NVs are most strongly relaxed at the instability cutoff. The NV response to the driven NZAFO magnetization can be understood using the schematic in Fig. 3c. At each magnetic field where the NZAFO shows microwave absorption a spinwave mode (black circle) will be excited above its thermal occupation level, either a nonzero wavevector mode or the uniform mode depending on the magnetic field. The four-magnon scattering process shown in Fig. 3d redistributes the excess population of magnons at the microwave drive frequency throughout the magnon spectrum. Four magnon scattering conserves both momentum and energy. A microwave excited magnon (black circle) scatters with a thermally occupied magnon (red circle), resulting in two product magnons (purple and yellow circles), some of which may be NV-resonant. An increase in NV-resonant magnon population as a result of this four-magnon scattering will result in a change in the NV PL level. Stray dipole magnetic fields from the NV-resonant magnons couple to and relax nearby NV centers. The relaxation of NVs shifts NV population out of the $m_s = 0$ spin state which we optically detect as a decrease in the ensemble NV PL.

Varying the microwave power allows us to shift the instability shoulder and therefore the instability-driven spinwave wavevector. Power dependent microwave absorption and NV relaxation signals at 2.2 GHz are given in Fig. 3e, f, respectively. We rule out temperature increase as a factor in the changing microwave absorption and NV relaxation signals in the Supplementary Figs. 1 and 2. In the microwave absorption data the instability shoulder shifts toward lower field with increasing power and the NV relaxation signal continues to be maximal at the lowest field

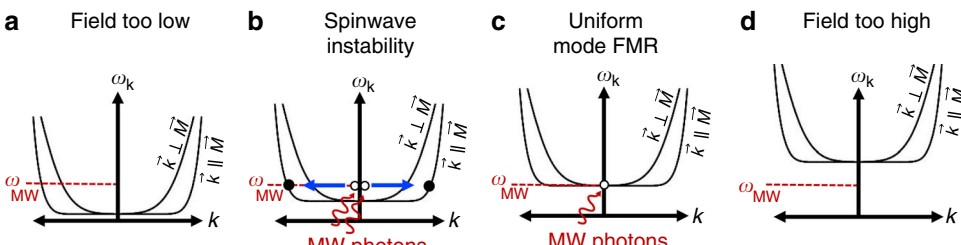

**Fig. 2 Microwave absorption processes during in-plane magnetic field sweep. a** When the magnetic field is well below the resonance condition microwaves do not efficiently couple to the uniform FMR mode and there is no microwave absorption. **b** When the magnetic field becomes close to the resonance condition, strong microwave drive can couple to the uniform mode FMR. In this process two microwave photons are absorbed at $\omega_{MW}$, giving two uniform mode magnons at $\omega_{MW}$ with $k = 0$ (white circles) which then decay to a counterpropagating pair of magnons with the same energy, but nonzero-$k$ (solid black circles). The microwave intensity threshold for this process is lowest for the $\vec{k} \parallel \vec{M}$ spinwave branch. The magnetic field range for which this process occurs is microwave power dependent, resulting in a lineshape change at high microwave powers. **c** On resonance, the typical microwave excitation of uniform mode precession occurs. **d** When the magnetic field exceeds the resonance condition there is no microwave absorption due to an absence of available magnon states at the microwave drive frequency.

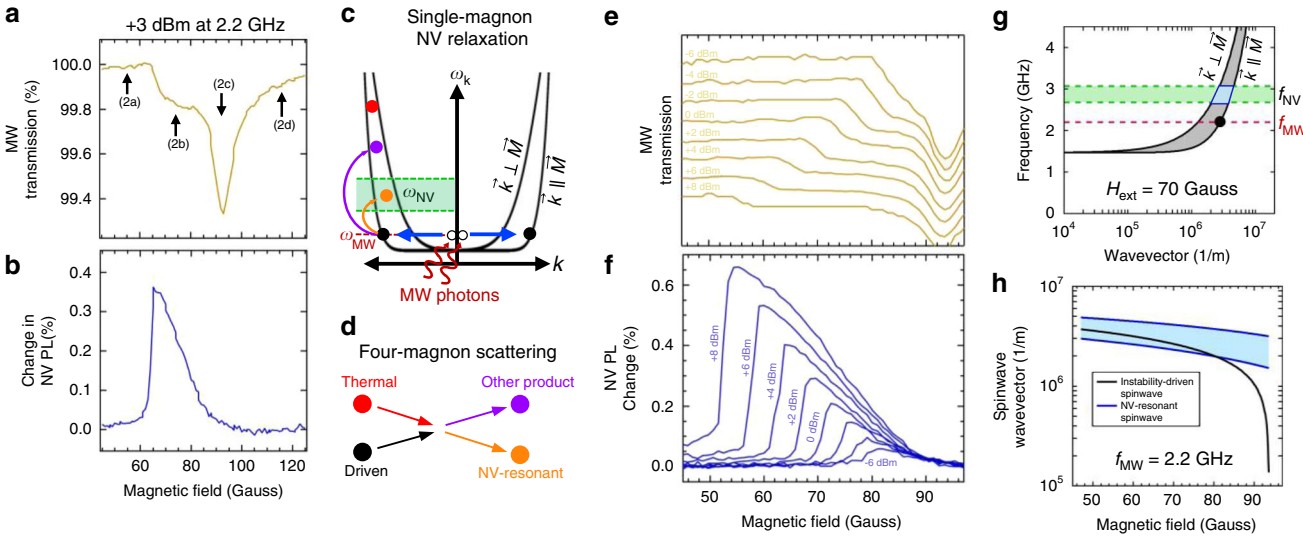

**Fig. 3 NV detection of spinwave instability in the single-magnon NV relaxation regime. a** Microwave absorption during a static magnetic field sweep at 2.2 GHz and +3 dBm microwave power shows a uniform mode FMR absorption signal at 94 Gauss and an absorption shoulder at lower magnetic field, consistent with the model in Fig. 2. **b** Simultaneously collected NV relaxation is strongest at the lowest field where the instability is driven and is minimal at the uniform mode FMR condition. **c** Out-of-equilibrium magnons driven by the instability (black) scatter off thermally occupied magnons (red), resulting in product magnons, some of which may be NV-resonant (yellow). An increase in population of NV-resonant magnons increases the dipole field noise at NV frequencies, leading to a change in NV PL. **d** The four-magnon scattering process redistributes excess magnon population from the driven mode to higher frequencies. **e** Increasing microwave power shifts the instability shoulder detected by microwave absorption toward lower magnetic field. Curves are vertically shifted from one another for readability. **f** NV relaxation signal follows the instability shoulder, shifting toward lower magnetic field with increasing microwave power. **g** Calculated spinwave manifold (gray) and NV center frequency range (green) at 70 Gauss gives the wavevector of the 2.2 GHz instability-driven magnon ($2.8 \times 10^6$ m$^{-1}$, black circle) and the range of wavevectors ($2.4 \times 10^6$ m$^{-1}$ to $4.2 \times 10^6$ m$^{-1}$, blue region) responsible for NV relaxation. **h** Calculated wavevector of 2.2 GHz instability-driven spinwave and NV-resonant spinwave wavevector range as a function of magnetic field.

for which the instability is driven. The NV relaxation signal increases with increasing driven spinwave wavevector, suggesting that the rate of the four-magnon scattering process responsible for populating the NV-resonant magnons increases with increasing wavevector of the instability-driven magnon. For each magnetic field of the experiment we calculate the spinwave spectrum and the range of NV ground state frequencies as shown in Fig. 3g (see Supplementary Discussion for spinwave spectrum calculation). From this we can determine the wavevector of the $\vec{\mathbf{k}} \parallel \vec{\mathbf{M}}$ spinwave excited by the instability and the wavevector range responsible for NV relaxation. At 70 Gauss the driven spinwave has $|\vec{\mathbf{k}}| = 2.8 \times 10^6$ m$^{-1}$ and the wavevector range of spinwaves responsible for NV relaxation is between 2.4 and 4.2 × $10^6$ m$^{-1}$, values which are well into the exchange branch of the spinwaves in NZAFO as can be seen in Fig. 3g. Performing these calculations across the range of magnetic fields where the instability can be driven in Fig. 3e, f we can determine the driven and detected spinwave wavevector values for the entire measurement range as given in Fig. 3h. In principle we can source higher microwave power in order to increase the driven and detected wavevector, however, at 50 Gauss the NZAFO spins are no longer saturated along the direction of the magnetic field due to the presence of a cubic anisotropy.

We then increase the microwave frequency and magnetic field range in order to lift the spinwave frequencies above the NV spin frequencies. Performing the same measurement as before at a microwave pump frequency of 4.0 GHz and +18 dBm microwave power we find that the NV is still relaxed by the driven instability even when there are no NV-resonant spinwave modes available. The microwave absorption in Fig. 4a shows a uniform mode FMR field of 206 Gauss and the low-field absorption feature of the instability with a cutoff near 175 Gauss. The NV response in Fig. 4b is largest at the instability cutoff, though the line shape of

the NV response is somewhat different than at 2.2 GHz. From the calculated spinwave spectrum at the magnetic field of the NV response peak, 175 Gauss, given in Fig. 4e we can see that there are no NV-resonant magnon modes in the film. Therefore the source of NV relaxation cannot be from single NV-resonant magnons as is the case at 2.2 GHz. A recent theoretical work by Flebus et al.[33]. showed that pairs of magnons can produce magnetic field noise at their difference frequency, which can lead to NV relaxation. A rough schematic of this mechanism is given in Fig. 4c, where the instability-excited spinwave mode (black circle) still undergoes the four-magnon scattering process in Fig. 4d, but now none of the four-magnon scattering products (purple circles) are themselves NV-resonant. One possibility, as Flebus et al[33]. have proposed, is that pairs of magnons whose frequencies differ by Δ can create magnetic field noise at this difference frequency, and if Δ = $f_{NV}$ this would lead to NV relaxation. Two propagating spinwaves of frequency difference Δ will modulate the local longitudinal magnetization as they move past a given region of the film which can result in a stray dipole field component at this frequency. Calculation of the spinwave spectrum at 4.0 GHz for the fields measured in our experiment allows us to extract the instability-excited wavevector for each value of the magnetic field in Fig. 4f.

Finally, we demonstrate the broadband applicability of our technique to measure NV relaxation for both one-magnon and multi-magnon NV resonant noise in Fig. 5. In each column, we plot the microwave absorption and NV response for a given nominal input microwave power. In order to remove artefacts from the microwave frequency response of our stripline we vary the input microwave power at each frequency in order to maintain a constant microwave transmission amplitude, which typically means that the listed power is sourced at 4.2 GHz and that at lower frequencies the input microwave power is somewhat less in

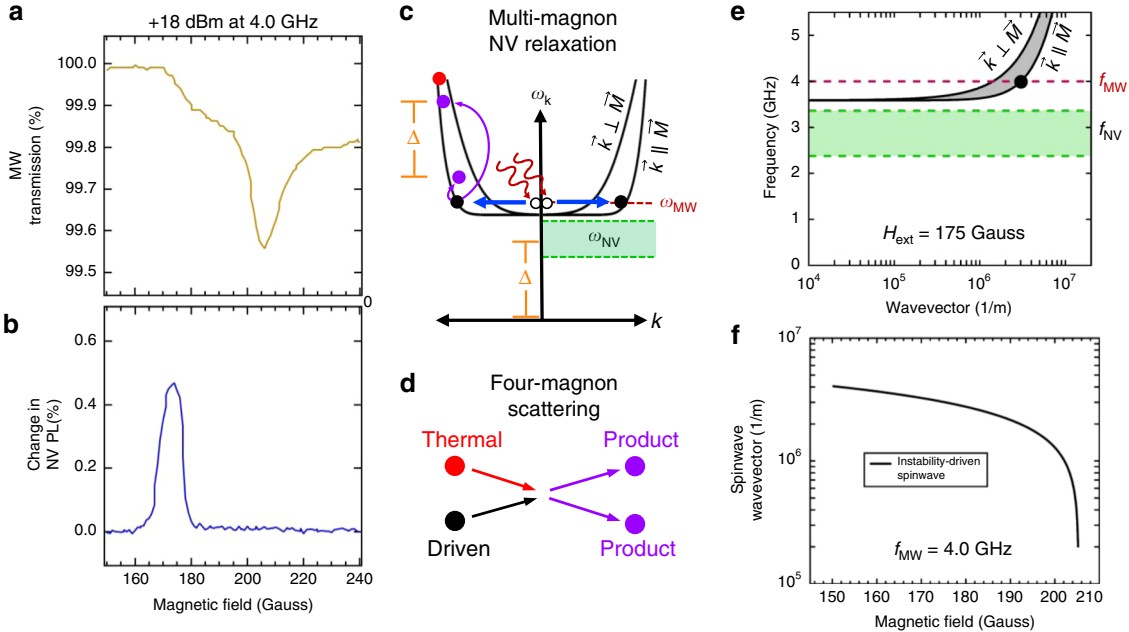

**Fig. 4 NV detection of spinwave instability in the multi-magnon NV relaxation regime. a** Microwave absorption during a static magnetic field sweep at 4.0 GHz and +18 dBm microwave power shows a uniform mode FMR absorption signal at 206 Gauss and an instability shoulder at lower field. **b** Simultaneously collected NV signal shows relaxation only at the lowest field where the instability is excited, and NV relaxation is minimal at the uniform mode FMR condition. **c** At these magnetic fields, the minimum of the spinwave manifold is lifted to higher frequency than the NV states, and there are no NV resonant magnons possible. An increase in NV-resonant magnetic field noise from multiple magnons generated in the four magnon scattering process (**d**) leads to a change in NV PL. **e** Calculated spinwave manifold (gray) and NV center frequency range (green) at 175 Gauss gives the wavevector of the 4.0 GHz instability-driven magnon ($3.2 \times 10^6$ m$^{-1}$, black circle). **f** Calculated wavevector of 4.0 GHz instability-driven spinwave as a function of magnetic field.

order to maintain flat transmission amplitude across the frequencies probed in the measurement. The input microwave power vs. frequency can be found in Supplementary Fig. 3. Figure 5a–c show the broadband evolution of the microwave absorption-detected spinwave instability and uniform mode FMR with increasing microwave power. Figure 5g–i show the broadband evolution of the NV relaxation signal with increasing microwave power. In Fig. 5d–f we plot the microwave absorption data with white circles labeling the microwave absorption-detected uniform mode FMR and with white triangles labeling the peak amplitude of the NV relaxation signal. It is clear that for all frequencies and fields the NV-ferromagnet coupling is strongest at the instability shoulder. As shown in Fig. 5j–l, we are able to separate the NV response to the ferromagnetic dynamics from the NV response to direct microwave absorption by calculating the resonance frequencies of NV ground and excited spin states oriented both parallel (solid red lines) and perpendicular (dashed red lines) at each field which fit the detected NV spectrum quite well. At magnetic fields below 50 Gauss the NZAFO demonstrates a resonant mode between 2.2 and 1.0 GHz which is a result of a cubic magnetic anisotropy that has been observed in previous studies[34] and is of little consequence for our results. Most importantly, we find that for high microwave power we are able to observe the instability-driven spinwaves at frequencies well in excess of the NV ground state transitions in Fig. 5i, l, a clear demonstration of broadband multi-magnon NV relaxometry. For comparison, we have included the PL spectrum of NV nanodiamonds on a non-magnetic substrate in Supplementary Fig. 4.

## Discussion
We have demonstrated NV detection of spinwaves generated via a nonlinear instability process in which spinwaves of nonzero wavevector are parametrically driven by a high amplitude

microwave field. The NV relaxation caused by these driven spinwaves can be divided into two regimes; one- and multi-magnon NV relaxometry. In the one-magnon relaxometry regime the driven spinwave frequency is below the NV frequency. The driven spinwave undergoes four-magnon scattering resulting in an increase in the population of NV-resonant magnons whose dipole fields couple to and relax the NV spins. In this one-magnon relaxometry regime the amplitude of the NV relaxation increases monotonically with the wavevector of the instability-excited spinwave mode. By varying the microwave drive power we are able to vary the driven spinwave wavevector up to $3 \times 10^6$ m$^{-1}$ which is well into the part of the spinwave spectrum dominated by the exchange interaction. In the case of multi-magnon NV relaxometry, the applied field is large enough that all spinwave frequencies are lifted to higher frequency than the NV frequencies. In this regime the NVs are relaxed by the driven spinwave instability despite the absence of individual NV-resonant magnons, suggesting that multiple magnons are creating noise below the ferromagnetic gap which couples to and relaxes the NV spins.

Further study should help to determine if the multi-magnon NV relaxation signal is caused by two-magnon processes or higher numbers of magnons producing NV-resonant noise, and which magnon wavevectors or frequencies are responsible for multi-magnon NV relaxation. Moreover, the two-magnon relaxation process was predicted to occur when the spin chemical potential of the system approaches the point of magnon Bose–Einstein condensation (BEC)[33]. Future work should include time-resolved studies of the NV relaxation rate like those by Du et al.[19] where it was shown that the NV can be used to directly measure the spin chemical potential. Quantifying the NV relaxation rate may also help to quantify how the wavevector of the instability-excited spinwave influences the four-magnon scattering rate. Time-resolved Brillouin light spectroscopy (BLS)

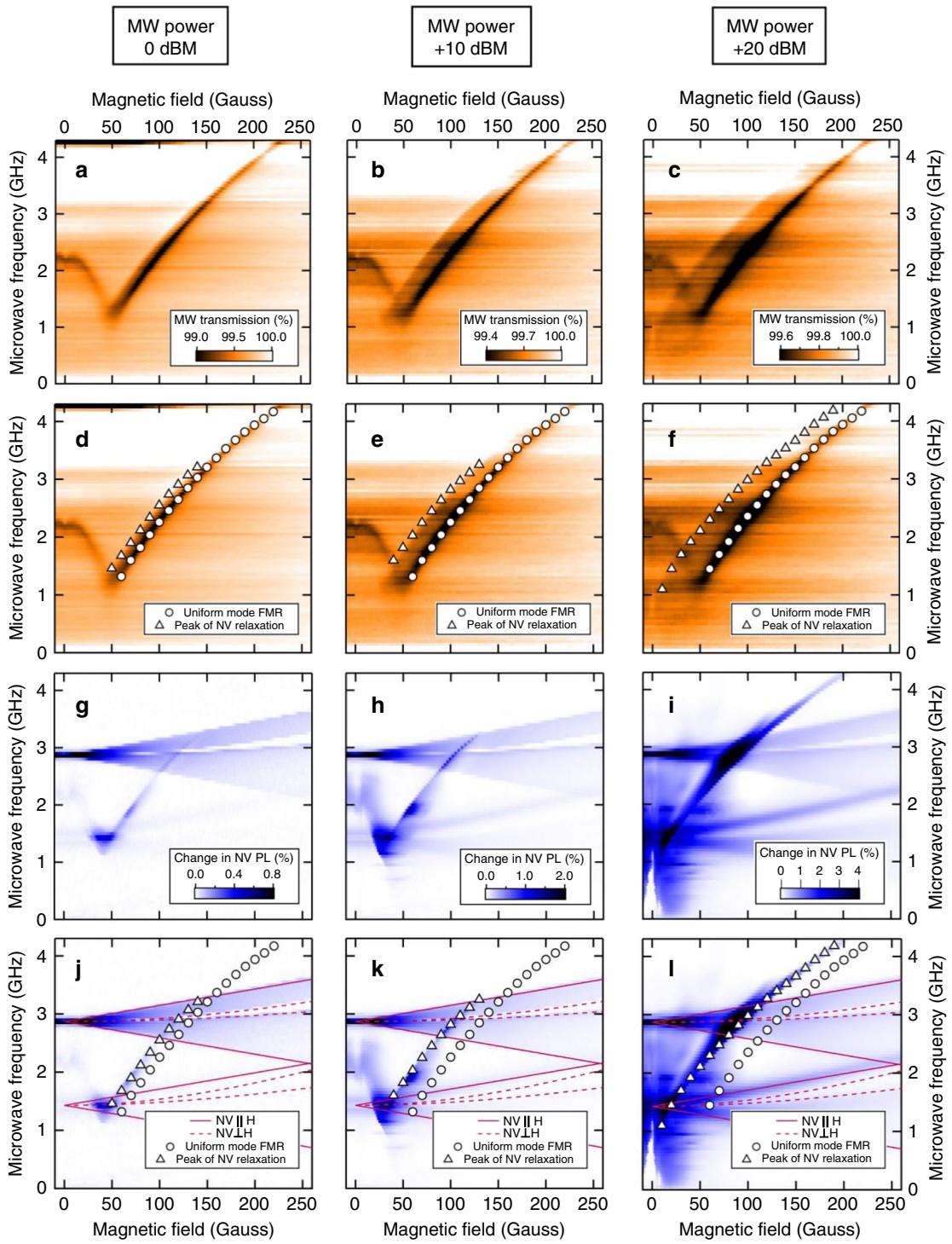

**Fig. 5 Broadband microwave absorption and NV relaxation signals.** Field-swept broadband microwave absorption at 0 dBm (**a**, **d**), +10 dBm (**b**, **e**), and +20 dBm (**c**, **f**). Input microwave amplitude at each frequency tuned to maintain constant microwave transmission amplitude off resonance (see Supplementary Fig. 3). Instability shoulder visible on the low-field side of the uniform mode FMR in **a**–**c**. NV relaxation signal at 0 dBm (**g**, **j**), +10 dBm (**h**, **k**), and +20 dbm (**i**, **l**) recorded simultaneously to corresponding microwave absorption data. White circles in panels **d**–**f**, **j**–**l** correspond to the microwave absorption-detected uniform mode FMR, and white triangles in these panels correspond to the maximum of the NV relaxation signal, showing that the NV always responds most strongly to the instability. Red lines in panels **j**–**l** are the calculated NV frequencies for the ground and excited NV states as a function of magnetic field for magnetic field oriented parallel to the applied field (solid) and perpendicular (dashed). The low-field ferromagnetic resonance feature below 50 Gauss arises from the cubic anisotropy field and is not of importance for our main results.

measurements[35] may help to elucidate if the NZAFO is driven to a magnon BEC for the geometry and microwave powers used in our experiment. The sharp cutoff of the NV signal at the onset of the spinwave instability suggests that NV relaxometry-based

imaging could be useful in understanding when nonlinear effects play a role in switching of ferromagnetic samples[36,37]. Lastly, the detection of ferromagnetic dynamics at above-NV frequencies should be generally useful for the magnonics community, where

the frequency of interest is often 10 or more GHz. Future work should determine if the decrease in the multi-magnon NV relaxometry signal with increasing microwave frequency and magnetic field (Fig. 5i, l) is intrinsic to the multi-magnon relaxometry process or if it is related to experimental constraints, like the quenching of NV contrast with increasing off-axis magnetic field[38] or decreasing wavevectors of spinwaves participating in the multi-magnon noise signal.

## Methods

**Lockin detection of NV and microwave absorption response.** Our measurement protocol is given in Fig. 1c. Conventional broadband microwave absorption and NV PL are measured simultaneously as the strength of the static magnetic field is swept and microwaves are applied at a constant frequency. In all, 100% amplitude modulated microwaves pass through the microstrip in transmission, are detected by a microwave diode, and measured at the modulation frequency by a lockin amplifier ($f_{mod} \sim 1$ kHz). The NV PL is detected by a photodiode and both the DC component of the PL is recorded as well as the changes in PL at the microwave modulation frequency by a second lockin. The phasing of the NV lockin is set such that a decrease in NV PL gives a positive lockin signal. NV signals are reported as percent changes in NV PL by dividing the lockin detected PL by the DC PL in order to remove some of the effects of the quenching of photoluminescence by off-axis oriented magnetic fields[38,39]. The microwave absorption signals are reported as percent transmission by normalizing with respect to the microwave transmission far from ferromagnetic absorption features. The lockin time constant used for all measurements was 100 ms.

## Data availability

The data supporting the findings of this study are available from the corresponding authors upon reasonable request.

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

## Acknowledgements

We thank Professor Phil Wigen for helpful discussion of spinwave instabilities. Funding for this research was provided primarily by the Center for Emergent Materials at The Ohio State University, a National Science Foundation (NSF) MRSEC through Award No. DMR-2011876, with partial support provided by the Air Force Office of Scientific Research (AFOSR) through Grants FA9550-19-1-0307 and FA9550-20RXCOR074. We acknowledge the use of Ohio State Nanosystems Laboratory (NSF DMR-2011876) shared facilities for device fabrication.

## Author contributions

B.A.M., A.M.T., V.P.B., M.R.P., and P.C.H. conceived the idea of the experiments. B.A.M., A.M.T., A.L.M. and M.R.P. acquired the data. B.A.M. and A.M.T. calculated spin-wave spectra and developed the theoretical understanding of the data with assistance from M.S.W., V.L.S., D.V.P., M.R.P., and P.C.H. B.A.G. grew the ferrite sample. B.A.M. and M.R.P. performed the microstrip antenna lithography. All authors discussed the results and participated in writing the paper.

## Competing interests

The authors declare no competing interests.
