## [Peer Review File · Nature Communications]

Reviewers' Comments:

Reviewer #1:

Remarks to the Author:

Dear Editor,

The manuscript "Ferromagnetic dynamics detected via one- and two-magnon NV relaxometry" present the experimental study of FMR under strong microwave driving, with varying pump power and bias magnetic field, and detected the dynamics through both the fluorescence of NV center and the microwave absorption spectra. The authors reported two regimes of NV response, one show instability, and the other one show two-magnon NV relaxometry. Although clear experimental results are provided, I found that the analysis is handwaving and may not be sufficient to support their conjecture. Additionally, it is not clear what is the innovation in this work and the potential impact of this work for the research fields.

Overall, This work is interesting and shows solid experimental results. I would recommend this work to Nature Communications after the authors after revising the theoretical analysis of this work.

Detailed comments:

(1) As mentioned in the introduction, "NV relaxometry ... has been employed as a sensitive tool for spectroscopic studies of ferromagnetic spinwaves [10, 13-19] ". It seems that the NV relaxometry has already been studied by other groups. So, what is new in this paper? For a publication in high impact journal as Nature Communications, there should be a new technique or new physics, or new phenomena to warrant its impact.

(2) In the paper, the authors claim that "Thus far NV relaxometry has been used to study ferromagnets with low damping since for low damping ferromagnets the driven ferromagnetic mode will undergo significantly more magnon-magnon scattering before eventually damping to vibrational modes." It is not clear what is the criteria of "low damping", and how "significantly" for the magnon-magnon scattering in the FMR.

(3) In Figure 1, when the magnetic field is around 50 Gauss, there is obvious fluorescence change of the NV by 1.4-1.5GHz driving. Is it related to the two-photon pumping to the NV? Because the NV has the transition frequencies around 2.9GHz.

(4) It is necessary to provide the spectra of NV fluorescence by directly drive the NV ensemble, without coupling to FMR. Then, we can compare the experimental results for the case with and without FMR, to conclude the effect of FMR.

(5) In Fig. 2(b), it is not clear what is the definition of "Area under FMR".

(6) For the second-order instability. What is the key feature of second-order instability? The authors just claim that "broadening of the FMR line and the shift" indicate the second-order instability. Could the author provide more quantitative evidence of the second-order instability? The authors claimed that "In the second-order instability process two microwave photons whose frequency is close to the uniform mode FMR frequency are converted into pairs of spinwaves." In their experiments, they drive the FMR with single microwave frequency, so please provide a schematic to show how it converted into pairs of spinwaves (show the momentum and energy conservations)?

(7) For NV Relaxometry Regime 2 (Fig. 4), it is not clear how the two-magnon process happens. Could the author provide a schematic or diagram to show the microwave photon and magnon creation/annihilation? For the results in Fig. 4(a), there is a peak of NV PL change at 170 Gauss, corresponding to an FMR frequency of ~ 3.5 GHz. So, how the energy conservation satisfied for the microwave drive at 4 GHz, FMR at 3.5 GHz and NV transition at 2.9 GHz?

Reviewer #2:

Remarks to the Author:

McCullian et al. measure the photoluminescence of nitrogen-vacancy defects in nanodiamonds that

are deposited on a thin film of the magnetic material "NZAFO". They do so while driving the NZAFO with microwave magnetic fields using a micro-strip line. The microwaves excite spin waves in the NZAFO that generate magnetic noise at the resonance frequencies of the NV defects, leading to a reduction in the NV photoluminescence as described in previous work. The main new observation in the current work is that such a reduction in the NV photoluminescence upon driving the ferromagnetic resonance can also occur when the ferromagnetic resonance frequency is larger than the NV frequencies, provided the microwave drive power is large enough. The authors attribute this observation to two-magnon processes as described in Ref. 24. The authors simultaneously measure the transmission of microwaves through the stripline in order to correlate these measurements to the observed features in the NV luminescence, attributing the shift and broadening in the inductively measured signal to a Suhl instability. The Suhl instability would then be responsible for the high magnon population necessary to generate large noise at energies in the spin-wave gap.

I find the paper not well written (there are linguistic errors resulting in incorrect statements, see example in detailed comments below) and the quality of the data low. Related to this, I find the arguments not well presented and difficult to follow. Most importantly, the data and arguments supporting the conclusion that the Suhl instability plays an important role in the observed response of the NV centers are too limited and also not well presented. This is related to that the amount of data is rather small: figure 1&2 present new information, but figure 3 and 4 are essentially linecuts of figure 1a at fixed frequency – only at microwave powers not yet shown in 1a. In summary the arguments should be formulated more carefully/presented more clearly, and the data should be more extensive with variation of parameters and of higher quality. I do not recommend this article to be accepted for publication.

Detailed comments:

- On page 2, it is stated that spin waves "shorten the effective NV spin relaxation rate". It seems the authors meant to say that the rate actually becomes larger.
- In Fig. 1a, a cutoff of the NV signal is present at low frequency and field (i.e. ~ 1 GHz, 40 G). The authors should comment of this effect.
- Figure 1c is referred as 1 "b" in its caption.
- In Fig. 2e, the discrepancy between the measured and calculated value seems to be of a factor ~ 2 for the threshold power (~ 6 dBm and ~ 3 dBm), which would increase to a factor above 3 for a field of 200 G (reading data off from Fig. 2a and Fig. 2c). The authors should comment on a possible explanation for this discrepancy and if possible analyse the data with a model to extract parameters? The explanation that this is a Suhl instability relies mostly on references/the arguments are not well presented. The statement that the comparison is only "... between the general trends" (from the last sentence of section 4 of the supplementary material) does not help in understanding the data.
- Since the authors measured the power transmission through the setup and tuned the power generated by the microwave source to keep the power at the sample constant, it would be beneficial to show such data.
- In figure 3a, the role of the colored bins of different height is unclear.
- For a more complete comparison of the PL and microwave transmission data, it would help to show the latter in their complete version (i.e. the full color-coded transmission vs frequency and field) and not only the maximum absorption value. Since they are measured simultaneously, this data should be already in possession of the authors. Particularly, it would be interesting to check whether the absorption signal resembles the signal of Fig. 1a.

Thursday, July 16, 2020

We thank the referees for their thoughtful reviews of our manuscript. We have made substantial changes in response to reviewer comments which we believe make the manuscript suitable for publication. Below we have included our response to the individual reviewer comments in red.

Response to Reviewer 1:

We are pleased to hear that the work is interesting with solid experimental results. We hope that the reviewer finds our modifications satisfactory in explaining the physics.

1. As mentioned in the introduction, "NV relaxometry ... has been employed as a sensitive tool for spectroscopic studies of ferromagnetic spinwaves [10, 13-19] ". It seems that the NV relaxometry has already been studied by other groups. So, what is new in this paper? For a publication in high impact journal as Nature Communications, there should be a new technique or new physics, or new phenomena to warrant its impact.

The new phenomena in our work is that we find that spinwaves can couple to and relax nearby NV centers *even when the NV frequency is in the ferromagnetic gap*. In all previous studies of NV-spinwave coupling the NV is relaxed by a single, NV-resonant spinwave mode, which limits the upper frequency threshold of NV-spinwave coupling to the NV frequency. Our results indicate that under high amplitude microwave drive the magnon population can be brought sufficiently far out of equilibrium that multiple magnons participate in producing magnetic field noise at frequencies below the ferromagnetic gap. This phenomenon was theoretically proposed in reference [35] of the main text; our work constitutes the first detection of this effect.

2. In the paper, the authors claim that "Thus far NV relaxometry has been used to study ferromagnets with low damping since for low damping ferromagnets the driven ferromagnetic mode will undergo significantly more magnon-magnon scattering before eventually damping to vibrational modes." It is not clear what is the criteria of "low damping", and how "significantly" for the magnon-magnon scattering in the FMR.

Our previous language was unclear. We have updated the text to better indicate the important physics: the generation of nonzero-k spinwaves via a spinwave instability process results in a spinwave population sufficiently out of equilibrium that NV-resonant magnetic field noise can be detected with nearby NVs.

3. In Figure 1, when the magnetic field is around 50 Gauss, there is obvious fluorescence change of the NV by 1.4-1.5GHz driving. Is it related to the two-photon pumping to the NV? Because the NV has the transition frequencies around 2.9GHz.

In our old Figure 1 (this data is now part of Figure 5 in the updated version) there are two sources of NV fluorescence change: direct ESR of the NV centers via the microwave field and relaxation of the NV spins via ferromagnetic dynamics. In Figure 5 of the updated version we have explicitly calculated and labelled with red lines the NV spin transition frequencies for both the NV ground state (2.87 GHz at zero applied field) and the NV excited state (1.40 GHz at zero applied field). The addition of these calculated NV ESR frequencies should help to disambiguate which NV responses result from direct ESR drive and which result from ferromagnetic dynamics.

4. It is necessary to provide the spectra of NV fluorescence by directly drive the NV ensemble, without coupling to FMR. Then, we can compare the experimental results for the case with and without FMR, to conclude the effect of FMR.

We have included the calculated NV ESR spectra to Figure 5 of the updated text. Additionally, we have included the fluorescence response spectrum of a nanodiamond ensemble deposited on an insulating non-magnetic substrate in the updated supplementary material for comparison with the data in Figure 5.

5. In Fig. 2(b), it is not clear what is the definition of "Area under FMR".

Area under FMR referred to the integrated area under the FMR peak. We have removed this figure from the updated version of the text.

6. For the second-order instability. What is the key feature of second-order instability? The authors just claim that "broadening of the FMR line and the shift" indicate the second-order instability. Could the author provide more quantitative evidence of the second-order instability? The authors claimed that "In the second-order instability process two microwave photons whose frequency is close to the uniform mode FMR frequency are converted into pairs of spinwaves." In their experiments, they drive the FMR with single microwave frequency, so please provide a schematic to show how it converted into pairs of spinwaves (show the momentum and energy conservations)?

We have included a schematic figure, Figure 2, which explains the physical mechanism of the second-order instability. Above a certain microwave drive threshold, a ferromagnetic film can be excited into uniform mode precession, near to but not on the uniform mode resonance condition. These uniform mode magnons then decay into magnons at the same frequency, but at nonzero-k. During a field-swept FMR measurement for a thin ferromagnetic film with the static field oriented in the film plane, this results in a microwave power dependent microwave absorption feature at the low field side of the FMR uniform mode resonance.

7. For NV Relaxometry Regime 2 (Fig. 4), it is not clear how the two-magnon process happens. Could the author provide a schematic or diagram to show the microwave photon and magnon creation/annihilation? For the results in Fig. 4(a), there is a peak of NV PL change at 170 Gauss, corresponding to an FMR frequency of ~3.5 GHz. So, how the energy conservation satisfied for the microwave drive at 4 GHz, FMR at 3.5 GHz and NV transition at 2.9 GHz?

In the theoretical proposal of two-magnon NV relaxometry, reference [35] of the updated text, the authors propose that as two magnons whose frequencies differ by Δ interact with the local film magnetization, that the longitudinal magnetization would be modulated at Δ . This modulation of the longitudinal magnetization could result in stray dipole fields at frequency Δ , supposing that this modulation of the longitudinal magnetization is inhomogeneous. Further study will be required to understand if this model sufficiently captures the physics of our observed relaxation signals.

We are exceedingly pleased to see that Reviewer #2's summary of the manuscript captures the core elements of our work so accurately. We take this as evidence that the aim of the work and the experimental methods are clear. Reviewer #2 has indicated concern over the writing clarity of the manuscript, the quality and breadth of the data as presented, and our argument that the Suhl instability is central to the physics presented. We have significantly re-written the manuscript to address all of these points, and feel that the revised version is of publishable clarity and quality.

1. On page 2, it is stated that spin waves “shorten the effective NV spin relaxation rate”. It seems the authors meant to say that the rate actually becomes larger.

We have updated the text on page 2 paragraph 2 to read “an increase in the number of NV-resonant magnons, which produce dipole magnetic field noise at the NV frequency and increase the relaxation rate of nearby NV spins”

2. In Fig. 1a, a cutoff of the NV signal is present at low frequency and field (i.e. ~1GHz, 40 G). The authors should comment of this effect.

We have updated the text on page 6 paragraph 1 to read “At magnetic fields below 50 Gauss the NZAFO demonstrates a resonant mode between 2.2 and 1.0 GHz which is a result of a cubic magnetic anisotropy that has been observed in previous studies”

3. Figure 1c is referred as 1 “b” in its caption.

We have re-arranged the figures and have addressed caption issues.

4. In Fig. 2e, the discrepancy between the measured and calculated value seems to be of a factor ~2 for the threshold power (~6 dBm and ~3 dBm), which would increase to a factor above 3 for a field of 200 G (reading data off from Fig. 2a and Fig. 2c). The authors should comment on a possible explanation for this discrepancy and if possible analyse the data with a model to extract parameters? The explanation that this is a Suhl instability relies mostly on references/the arguments are not well presented. The statement that the comparison is only “... between the general trends” (from the last sentence of section 4 of the supplementary material) does not help in understanding the data.

We have removed what was formerly Figure 2 from the paper. Instead we have included power dependent microwave absorption and NV relaxation data in the new Figure 3 (panels E and F) which show explicitly the evolution of the microwave absorption shoulder and NV response to this shoulder as a function of microwave power. Our new Figure 2 schematically explains the field-swept FMR lineshape when a sample undergoes the second-order spinwave instability, which is in agreement with the power dependent data in Figure 3.

5. Since the authors measured the power transmission through the setup and tuned the power generated by the microwave source to keep the power at the sample constant, it would be beneficial to show such data.

We have added the microwave transmission data to Figure 3 (A and E), Figure 4 (A), and Figure 5 (A through F). We thank the reviewer especially for this suggestion. The relaxation of the NV being maximum at the microwave power dependent microwave absorption shoulder is very strong evidence for our explanation that a spinwave instability followed by magnon-magnon scattering results in the strongest NV relaxation.

6. In figure 3a, the role of the colored bins of different height is unclear.

We have removed the colored bins. The bins were to line up different parts of the NV and microwave absorption profiles to the schematic below in that figure. In the new figures, particularly in Figure 3 (a) we have added four indicators (2a) (2b) (2c) and (2d) which point the reader back to the relevant sub-figure of Figure 2 where the absorption profile is schematically explained. We hope that this provides some clarity in interpreting the microwave absorption profile.

7. For a more complete comparison of the PL and microwave transmission data, it would help to show the latter in their complete version (i.e. the full color-coded transmission vs frequency and field) and not only the maximum absorption value. Since they are measured simultaneously, this data should be already in possession of the authors. Particularly, it would be interesting to check whether the absorption signal resembles the signal of Fig. 1a

We again thank the reviewer for this particularly helpful insight. We now present both the NV and the microwave absorption broadband data in Figure 5, finding excellent agreement with the NV relaxation signal being maximal at the microwave absorption shoulder (but not at the microwave absorption detected uniform mode).

Reviewers' Comments:

Reviewer #1:

Remarks to the Author:

The authors significantly revised their manuscript, and now they provide more clear physical picture and discussions about their experimental results. Therefore, most of my concerns are addressed, and I would recommend its publication in Nature Communications.

Reviewer #2:

Remarks to the Author:

The authors have satisfactorily addressed all my comments. I think the work presents a major advance in the understanding of the magnetic fluctuations and instabilities of ferromagnets and in the application of NV centers to magnetism. I recommend publication.

Monday, August 31, 2020

We thank the referees for their thoughtful reviews of our manuscript. We have made substantial changes in response to reviewer comments which we believe make the manuscript suitable for publication. Below we have included our response to the first round of individual reviewer comments in **red**, and to the second round of reviewer comments in **blue**.

Response to Reviewer 1's first round of comments:

We are pleased to hear that the work is interesting with solid experimental results. We hope that the reviewer finds our modifications satisfactory in explaining the physics.

1. As mentioned in the introduction, "NV relaxometry ... has been employed as a sensitive tool for spectroscopic studies of ferromagnetic spinwaves [10, 13-19] ". It seems that the NV relaxometry has already been studied by other groups. So, what is new in this paper? For a publication in high impact journal as Nature Communications, there should be a new technique or new physics, or new phenomena to warrant its impact.

The new phenomena in our work is that we find that spinwaves can couple to and relax nearby NV centers *even when the NV frequency is in the ferromagnetic gap*. In all previous studies of NV-spinwave coupling the NV is relaxed by a single, NV-resonant spinwave mode, which limits the upper frequency threshold of NV-spinwave coupling to the NV frequency. Our results indicate that under high amplitude microwave drive the magnon population can be brought sufficiently far out of equilibrium that multiple magnons participate in producing magnetic field noise at frequencies below the ferromagnetic gap. This phenomenon was theoretically proposed in reference [35] of the main text; our work constitutes the first detection of this effect.

2. In the paper, the authors claim that "Thus far NV relaxometry has been used to study ferromagnets with low damping since for low damping ferromagnets the driven ferromagnetic mode will undergo significantly more magnon-magnon scattering before eventually damping to vibrational modes." It is not clear what is the criteria of "low damping", and how "significantly" for the magnon-magnon scattering in the FMR.

Our previous language was unclear. We have updated the text to better indicate the important physics: the generation of nonzero-k spinwaves via a spinwave instability process results in a spinwave population sufficiently out of equilibrium that NV-resonant magnetic field noise can be detected with nearby NVs.

3. In Figure 1, when the magnetic field is around 50 Gauss, there is obvious fluorescence change of the NV by 1.4-1.5GHz driving. Is it related to the two-photon pumping to the NV? Because the NV has the transition frequencies around 2.9GHz.

In our old Figure 1 (this data is now part of Figure 5 in the updated version) there are two sources of NV fluorescence change: direct ESR of the NV centers via the microwave field and relaxation of the NV spins via ferromagnetic dynamics. In Figure 5 of the updated version we have explicitly calculated and labelled with red lines the NV spin transition frequencies for both the NV ground state (2.87 GHz at zero applied field) and the NV excited state (1.40 GHz at zero applied field). The addition of these calculated NV ESR frequencies should help to disambiguate which NV responses result from direct ESR drive and which result from ferromagnetic dynamics.

4. It is necessary to provide the spectra of NV fluorescence by directly drive the NV ensemble, without coupling to FMR. Then, we can compare the experimental results for the case with and without FMR, to conclude the effect of FMR.

We have included the calculated NV ESR spectra to Figure 5 of the updated text. Additionally, we have included the fluorescence response spectrum of a nanodiamond ensemble deposited on an insulating non-magnetic substrate in the updated supplementary material for comparison with the data in Figure 5.

5. In Fig. 2(b), it is not clear what is the definition of "Area under FMR".

Area under FMR referred to the integrated area under the FMR peak. We have removed this figure from the updated version of the text.

6. For the second-order instability. What is the key feature of second-order instability? The authors just claim that "broadening of the FMR line and the shift" indicate the second-order instability. Could the author provide more quantitative evidence of the second-order instability? The authors claimed that "In the second-order instability process two microwave photons whose frequency is close to the uniform mode FMR frequency are converted into pairs of spinwaves." In their experiments, they drive the FMR with single microwave frequency, so please provide a schematic to show how it converted into pairs of spinwaves (show the momentum and energy conservations)?

We have included a schematic figure, Figure 2, which explains the physical mechanism of the second-order instability. Above a certain microwave drive threshold, a ferromagnetic film can be excited into uniform mode precession, near to but not on the uniform mode resonance condition. These uniform mode magnons then decay into magnons at the same frequency, but at nonzero-k. During a field-swept FMR measurement for a thin ferromagnetic film with the static field oriented in the film plane, this results in a microwave power dependent microwave absorption feature at the low field side of the FMR uniform mode resonance.

7. For NV Relaxometry Regime 2 (Fig. 4), it is not clear how the two-magnon process happens. Could the author provide a schematic or diagram to show the microwave photon and magnon creation/annihilation? For the results in Fig. 4(a), there is a peak of NV PL change at 170 Gauss, corresponding to an FMR frequency of ~3.5 GHz. So, how the energy conservation satisfied for the microwave drive at 4 GHz, FMR at 3.5 GHz and NV transition at 2.9 GHz?

In the theoretical proposal of two-magnon NV relaxometry, reference [35] of the updated text, the authors propose that as two magnons whose frequencies differ by Δ interact with the local film magnetization, that the longitudinal magnetization would be modulated at Δ . This modulation of the longitudinal magnetization could result in stray dipole fields at frequency Δ , supposing that this modulation of the longitudinal magnetization is inhomogeneous. Further study will be required to understand if this model sufficiently captures the physics of our observed relaxation signals.

Response to Reviewer 1's second round of comments:

1. The authors significantly revised their manuscript, and now they provide more clear physical picture and discussions about their experimental results. Therefore, most of my concerns are addressed, and I would recommend its publication in Nature Communications.

We are pleased to hear that our first round of revisions addressed the concerns raised by Reviewer #1.

Response to Reviewer 2's first round of comments:

We are exceedingly pleased to see that Reviewer #2's summary of the manuscript captures the core elements of our work so accurately. We take this as evidence that the aim of the work and the experimental methods are clear. Reviewer #2 has indicated concern over the writing clarity of the manuscript, the quality and breadth of the data as presented, and our argument that the Suhl instability is central to the physics presented. We have significantly re-written the manuscript to address all of these points, and feel that the revised version is of publishable clarity and quality.

1. On page 2, it is stated that spin waves "shorten the effective NV spin relaxation rate". It seems the authors meant to say that the rate actually becomes larger.

We have updated the text on page 2 paragraph 2 to read "an increase in the number of NV-resonant magnons, which produce dipole magnetic field noise at the NV frequency and increase the relaxation rate of nearby NV spins"

2. In Fig. 1a, a cutoff of the NV signal is present at low frequency and field (i.e. ~1GHz, 40 G). The authors should comment of this effect.

We have updated the text on page 6 paragraph 1 to read "At magnetic fields below 50 Gauss the NZAFO demonstrates a resonant mode between 2.2 and 1.0 GHz which is a result of a cubic magnetic anisotropy that has been observed in previous studies"

3. Figure 1c is referred as 1 "b" in its caption.

We have re-arranged the figures and have addressed caption issues.

4. In Fig. 2e, the discrepancy between the measured and calculated value seems to be of a factor ~2 for the threshold power (~6 dBm and ~3 dBm), which would increase to a factor above 3 for a field of 200 G (reading data off from Fig. 2a and Fig. 2c). The authors should comment on a possible explanation for this discrepancy and if possible analyse the data with a model to extract parameters? The explanation that this is a Suhl instability relies mostly on references/the arguments are not well presented. The statement that the comparison is only "... between the general trends" (from the last sentence of section 4 of the supplementary material) does not help in understanding the data.

We have removed what was formerly Figure 2 from the paper. Instead we have included power dependent microwave absorption and NV relaxation data in the new Figure 3 (panels E and F) which show explicitly the evolution of the microwave absorption shoulder and NV response to this shoulder as a function of microwave power. Our new Figure 2 schematically explains the field-swept FMR lineshape when a sample undergoes the second-order spinwave instability, which is in agreement with the power dependent data in Figure 3.

5. Since the authors measured the power transmission through the setup and tuned the power generated by the microwave source to keep the power at the sample constant, it would be beneficial to show such data.

We have added the microwave transmission data to Figure 3 (A and E), Figure 4 (A), and Figure 5 (A through F). We thank the reviewer especially for this suggestion. The relaxation of the NV being maximum at the microwave power dependent microwave absorption shoulder is very strong evidence for our explanation that a spinwave instability followed by magnon-magnon scattering results in the strongest NV relaxation.

6. In figure 3a, the role of the colored bins of different height is unclear.

We have removed the colored bins. The bins were to line up different parts of the NV and microwave absorption profiles to the schematic below in that figure. In the new figures, particularly in Figure 3 (a) we have added four indicators (2a) (2b) (2c) and (2d) which point the reader back to the relevant sub-figure of Figure 2 where the absorption profile is schematically explained. We hope that this provides some clarity in interpreting the microwave absorption profile.

7. For a more complete comparison of the PL and microwave transmission data, it would help to show the latter in their complete version (i.e. the full color-coded transmission vs frequency and field) and not only the maximum absorption value. Since they are measured simultaneously, this data should be already in possession of the authors. Particularly, it would be interesting to check whether the absorption signal resembles the signal of Fig. 1a

We again thank the reviewer for this particularly helpful insight. We now present both the NV and the microwave absorption broadband data in Figure 5, finding excellent agreement with the NV relaxation signal being maximal at the microwave absorption shoulder (but not at the microwave absorption detected uniform mode).

Response to Reviewer 2's second round of comments:

1. The authors have satisfactorily addressed all my comments. I think the work presents a major advance in the understanding of the magnetic fluctuations and instabilities of ferromagnets and in the application of NV centers to magnetism. I recommend publication.

We are pleased to learn that our first round of revisions addressed the concerns raised by Reviewer #1, and are grateful for the kind words about our manuscript.